# Gold-Polymer Nanocomposites for Future Therapeutic and Tissue Engineering Applications

**DOI:** 10.3390/pharmaceutics14010070

**Published:** 2021-12-28

**Authors:** Panangattukara Prabhakaran Praveen Kumar, Dong-Kwon Lim

**Affiliations:** 1KU-KIST Graduate School of Converging Science and Technology, Korea University, 145 Anam-ro, Seongbuk-gu, Seoul 02841, Korea; p4praveen.18@gmail.com; 2Department of Integrative Energy Engineering, Korea University, 145 Anam-ro, Seongbuk-gu, Seoul 02841, Korea

**Keywords:** gold nanoparticles, polymers, drug delivery, tissue engineering, photothermal therapy, photodynamic therapy

## Abstract

Gold nanoparticles (AuNPs) have been extensively investigated for their use in various biomedical applications. Owing to their biocompatibility, simple surface modifications, and electrical and unique optical properties, AuNPs are considered promising nanomaterials for use in in vitro disease diagnosis, in vivo imaging, drug delivery, and tissue engineering applications. The functionality of AuNPs may be further expanded by producing hybrid nanocomposites with polymers that provide additional functions, responsiveness, and improved biocompatibility. Polymers may deliver large quantities of drugs or genes in therapeutic applications. A polymer alters the surface charges of AuNPs to improve or modulate cellular uptake efficiency and their biodistribution in the body. Furthermore, designing the functionality of nanocomposites to respond to an endo- or exogenous stimulus, such as pH, enzymes, or light, may facilitate the development of novel therapeutic applications. In this review, we focus on the recent progress in the use of AuNPs and Au-polymer nanocomposites in therapeutic applications such as drug or gene delivery, photothermal therapy, and tissue engineering.

## 1. Introduction

In ancient times, gold (Au) was used to treat various diseases, including fever, syphilis, tuberculosis, and rheumatism [1]. It exists in the diverse oxidation states (−I), (0), (I), (II), (III), (IV), and (V); however, only Au(0), Au(I), and Au(III) are stable in an aqueous solution. Au(I) is highly water-soluble, may be easily stabilized by ligands, and was used as a therapeutic agent in injectable and oral preparations, but later studies revealed the toxicity of these formulations. The most common side effects associated with Au treatment are skin and mucous membrane hypersensitivity reactions and allergies [2]. Conversely, Au(III) is very rarely used as a therapeutic agent due to its strong oxidizing nature. Examples of Au complexes used in clinical practices are Aurolate [3] and Auranofin [4], which are used in psoriatic and rheumatoid arthritis and oral cavity treatment, respectively (Figure 1) [1].

In contrast, Au nanoparticles (AuNPs) are nanosized particles prepared via the reduction of Au(I) or Au(III) to Au(0), with toxicities that differ remarkably from those of Au salts. The properties of AuNPs also differ considerably from those of bulk Au metal, which is bright yellow and extremely inert. However, the colloidal Au displays a bright red-wine color (Figure 1). The toxicity of colloidal Au is higher than that of bulk Au, but the toxicity of colloidal Au is significantly lower than that of Au salts. Hence, AuNPs exhibit completely different biological behaviors in terms of biodistribution, cellular uptake efficiency, and therapeutic outcomes. Accordingly, the application of AuNPs in medicine differs significantly from that of the previously reported Au salts (Figure 1).

AuNPs exhibit strong light absorption and scattering properties in the visible and near-infrared (NIR) regions, owing to strong surface plasmon resonances at these wavelengths [5]. Based on their strong scattering properties, AuNPs were used as contrast-enhancing agents in electron microscopes (Figure 1) [6,7]. Thereafter, diverse nanostructures, such as Au nanospheres (AuNPs) [8], -rods (AuNRs) [9], -shells (AuNSs) [10], -stars (AuSTs) [11], -cages (AuCGs) [12], and -clusters (AuNCs) [13], were prepared and evaluated for their use in various applications in the field of nanomedicine. Along with their unique optical properties, the biocompatibility and diverse surface chemistry—covalent or non-covalent—of AuNPs facilitated their use in various biomedical applications as novel tools in biosensing and -imaging and therapeutics (Figure 1) [14,15]. 

Diverse surface chemistry is required in various applications. Among the strategies for surface modification of AuNPs, modification with polymers yielded considerably improved performances in biomedical applications, particularly as drug delivery and tissue engineering platforms [16,17,18]. Polymers can enhance stability and dispersity and further expand the functionality of AuNPs [19,20]. Polymer shells promote the preservation of the optical properties of the Au core and induce responses to endo- or exogenous stimuli, such as enzymes, pH, temperature, or light [21,22]. For therapeutic application, high loading and controlled release of active pharmaceuticals from the nanocarrier is critical and may be modulated using polymers. Extended circulation of the nanocarrier in the bloodstream is another key issue in designing an efficient drug delivery system. Functionalization with polyethylene glycol (PEG), referred to as PEGylation, can extend the circulation of AuNPs in the blood by minimizing opsonization by the immune system. For tissue engineering, extended retention and controlled release of active molecules are critical in achieving the intended functions [23,24]. In an active system, Au may mediate the external light or electrical stimulus of the biological system, in addition to improving the mechanical strengths of the scaffolds [25,26]. In this review, we focus on the recent progress regarding Au and Au-polymer hybrid nanomaterials that were developed for therapeutic and tissue engineering applications in nanomedicine.

## 2. Synthesis of Gold or Gold-Polymer Nanoparticles

The most common method used for the synthesis of AuNPs is the bottom-up chemical approach method. In each of the synthetic methods, precise control over the shape, size, and stability of the final AuNPs is taken care of. The first method for the synthesis of colloidal AuNPs is considered to be the method by Faraday, in which NaAuCl_4_ is reduced using phosphorous/carbon disulfide solution to obtain a ruby-colored colloidal solution of AuNPs with an average size of particles 6 ± 2 nm [27]. The hydrothermal method, in which sodium citrate was used for reducing HAuCl_4_, under reflux conditions, in which sodium citrate acts as a reducing and stabilizing agent, was later used; this method is known as the Turkevich method, which produced AuNPs with good dispersity [28].

Chemical reduction of HAuCl_4_ using NaBH_4_ as a reducing agent is another method for synthesizing AuNPs with an average size of 3 nm with longer time stability [29]. Seed mediated multistep method is also followed to synthesize AuNPs with different size and shape [30,31]. In this method, HAuCl_4_ is mixed with cetyltrimethyl ammonium bromide (CTAB) to form the Au(III)-CTA complex, followed by the reduction of Au(III) using NaBH_4_ or citrate to synthesize Au seeds. In the final step, various concentrations of Au seed were added to a growth solution containing AgNO_3_ and ascorbic acid to form nanostructures with different shapes, such as AuNPs, AuNRs, AuNSs, AuSTs, AuCGs, etc. [32]. Likewise, in recent years, interfacial synthesis [33], laser ablation [34], arc discharge [35], microwave [36], electrochemical [37], and biological and green chemistry methods [38,39,40] have also been investigated for the synthesis of AuNPs.

The stability and toxicity of the bare AuNPs is the most challenging issue in using them directly for various biomedical applications. The stabilization of AuNPs can be performed by either electrostatic or steric stabilization to avoid the aggregation of bare AuNPs, and this can be achieved by using surfactants or functionalized polymers to prepare Au-polymer nanocomposites.

Typically, polymers have an opposite charge to that of AuNPs that were physisorbed on the surface of AuNPs to minimize the aggregation tendency by creating a passive layer [41]. In the direct synthesis approach, reduction of HAuCl_4_ in the presence of sulfur or amino terminated polymers is carried out to obtain the hybrid polymer AuNPs in a one-step process [42,43]. Presently, the covalent bond formation between AuNPs and various polymers is employed for the successful synthesis of various Au-polymer nanocomposites. The covalent approach falls into the following categories: (i) graft-from approach, (ii) graft-to approach, and (iii) grafting-through technique, as shown in Figure 2 [44]. In the graft-from approach, AuNPs are first immobilized with the polymerization initiators (e.g., chain transfer agents) and, subsequently, the polymer chains grew by the addition of monomers, commonly known as the reverse addition-fragmentation chain transfer (RAFT) process. Takara et al. synthesized glycopolymers of polyacrylamide derivatives with mannose via the RAFT process, reducing the polymer terminal group to thiol, and prepared polymer-coated AuNPs with an average diameter of 55–70 nm, used for the mannose-protein interaction studies [45]. Otherwise, in the graft-to polymer method, the polymers with end functionalities, such as thiols, amines, etc., were covalently linked to the surface functionalities on AuNPs’ surface. In the grafting-through approach, the polymerizable groups are anchored on the surface of AuNPs and the polymerization starts in the solution containing the monomer, initiator, and the modified NPs, which act as a cross-linking agent [46]. Suzuki et al. reported the use of thiol and amino-functionalized polymer, poly(*N*-isopropyl acrylamide)-b-(glycidyl methacrylate), to stabilize AuNPs inside the polymer shells via the graft-to method to synthesize AuNPs with an average size of 10 nm [47].

Post-modification can also be performed, in which the as such prepared AuNPs are conjugated with the polymer, but this method lacks the low polymer loading efficiency and unintended adsorption through polymer functionalities. Mostly biopolymers, DNA, etc., are attached to the surface of AuNPs by post-modification methods [43,48]. For the potential application in cancer therapy, the size of polymer AuNPs is very crucial, and polymers such as polystyrene, polyethylene imine, poly(acryloyl amino-phenyl arsonic acid), and xanthan gum are used as stabilizing and reducing agents for the Au(III) [49]. The size distribution of AuNPs can be varied by using different molar mass polyethylene imine polymers and by varying the number of branching units, as demonstrated by Cho et al. [50].

## 3. Therapeutic Applications

NPs with sizes < 100 nm were widely investigated to overcome the limitations of chemotherapy, such as severe side effects of cytotoxic drugs, low maximum tolerable doses, and multidrug resistance. Among the numerous nanomaterials investigated for use in drug delivery systems, including polymeric micelles, polypeptides, and liposomes, AuNPs exhibit unique optical properties, such as strong light scattering at visible and NIR wavelengths, biocompatibility, and robust surface chemistry, which have accelerated their study in diverse fields of nanomedicine as passive carriers or active vehicles responding to light or electrical stimulus. In this section, we discuss the recent progress in drug delivery systems for therapeutic applications, such as cancer therapy, gene delivery, photothermal and -dynamic therapies, and cancer immunotherapy.

### 3.1. Drug Delivery

Compared with organic nanomaterials, AuNPs are inorganic nanomaterials that are not decomposed, even at low concentrations. They are rigid, stable nanomaterials, and the drug molecules may be loaded chemically or physically on AuNP surfaces. Cellular uptake of AuNPs may occur via several mechanisms, depending on the sizes, shapes, surface charges, and ligand functionality of the AuNPs [51]. The typical sizes of 10–100 nm were widely investigated, and numerous studies reported the size- and shape-dependent half-lives of AuNPs in the blood [52]. Polymer layers were easily introduced on AuNPs, which enhanced their drug-loading efficiency and circulation in the blood, changed their biodistribution, and added functionality [44].

AuNPs conjugated with anti-cancer drugs via electrostatic interactions or diverse linkers, including thiol, amide, or hydrazine, were utilized as carriers of these drugs [53]. The covalent linkers between AuNPs and the drugs could be designed in such a way that they are cleaved by a cancer cell-specific enzyme or low pH to efficiently escape the endosomes. However, the carrier initially overcomes the defense mechanism of the human body, the reticuloendothelial system, which is the primary barrier in the systematic delivery system. PEG is the most widely used polymer to overcome this barrier, as it absorbs numerous water molecules, which enables it to avoid recognition by macrophages. PEGylation not only avoids the removal of the AuNPs and nonspecific binding of the carrier during delivery but also improves the efficiency of drug delivery to the target cells.

For example, Yoon et al. proposed controlling drug release by changing the density of PEG on the AuNP carrier. A low density of PEG results in the rapid release of drugs, whereas a high density of PEG results in the slow release of methotrexate (MTX) due to protection by glutathione, as shown in Figure 3A,B [54]. Studies showed an increase in PEG density resulting in low drug loading and anticancer activity, suggesting a grafting density of 0.0011/nm^2^ PEG would be suitable for anticancer activity in HeLa cells. In vivo studies showed that a 50% surface PEGylation could induce prolonged blood circulation and tumor accumulation of 2 nm AuNPs [55].

In addition, Ghorbai et al. showed that multiple anti-cancer drugs may be loaded simultaneously onto PEGylated AuNPs, with high tumor-targeting specificity. Doxorubicin (DOX) and MTX were incorporated into AuNPs via ionic interactions, whereas 6-mercaptopurine was incorporated into AuNPs via disulfide bonds with the polymeric shell [56]. DOX and MTX release occurred via changes in pH, whereas 6-mercaptopurine was released via glutathione interaction. This study showed the stimuli-responsive release of drugs from the carrier. Farooq et al. reported PEG-coated AuNPs that could simultaneously deliver bleomycin and DOX into the HeLa cancer cell line [57]. Meena et al. prepared a pH-sensitive drug delivery system using PEG-coated AuNPs to simultaneously deliver DOX and kaempferol for efficient colon cancer therapy [58]. Drug release could be controlled using pH, and it exhibited an efficient cytotoxic effect compared to that of the individual drugs. A significant reduction in tumor volume was observed without significant side effects, highlighting the utility of the dual drug encapsulating nanocarrier as a good candidate for treating colon cancer. These studies showed the significance of PEG in improving the biocompatibility, non-immunogenicity, and dispersity and reducing the enzymatic degradation of NPs in the biological environment.

Various other biocompatible polymers were also utilized in drug delivery systems. For example, poly(sodium 4-styrenesulfonate) and AuNRs exhibited good biocompatibility and sustained release of DOX [59]. Chitosan (CS)-oligosaccharide-stabilized AuNPs were used in the controlled release of paclitaxel. The CS-oligosaccharide backbone improved the solubility and uptake efficiency of paclitaxel in MDA-MB-231 breast cancer cell lines. The nanocomposite displayed pH-dependent drug release in the endosome owing to the repulsion between the drugs and the protonated amino groups of CS [60]. Suarasan et al. reported gelatin-coated AuNPs as a controlled-release system for DOX. Their performance was examined in vitro using MCF-7 breast cancer cells at different pH values (7.4, 5.3, and 4.6) and temperatures (22–45 °C). At high temperature (45 °C) and low pH (4.6), enhanced release and the internalization of DOX were studied using time-resolved fluorescence and surface-enhanced Raman scattering of the aggregated AuNPs [61]. Wu et al. reported a NIR laser-triggered shrinkable formulation with poly(acrylamide-acrylonitrile)-PEG-lipoic acid encapsulating AuNRs [62]. During laser irradiation, the AuNRs convert light to heat energy, and the large micelles split into ultrasmall micelles (7 nm) that penetrate the tumor cells deeply to achieve the in situ controlled release of the loaded drug (DOX), as shown in Figure 3C. The reduction in the size of the tumor was monitored by incorporating a fluorescent imaging agent (Figure 3D). Other than fluorescent dyes, the incorporation of inorganic imaging agents, such as iron oxide or radioisotopes, is also extensively investigated, as this enables monitoring of the disease state or the tracking of the cell or agent in the body [63,64].

The rational design of stimuli-responsive Au-polymer nanocarriers for drug delivery and chemotherapy in the NIR region resulted in promising results. Lian et al. synthesized a coumarin derivative; 7-(diethylamino)-4-(hydroxymethyl)-2H-chromen-2-one (DEACM); which then immobilized on cyclodextrin (β-CDs) bearing AuNRs. The choice of coumarin derivative is meant to enhance the drug loading capacity of DOX via the µ-µ stacking interactions. PEG functionalized with cycle RGD peptide was linked to β-CDs for active targeting and long circulation process. The NIR light irradiation activated the gold nanorods and accelerated the photosolvolysis of DEACM to trigger the rapid release of DOX. In vitro studies showed the inhibition effect on the proliferation of 4T1 breast cancer cells, and in vivo studies for breast cancer in mice showed excellent anticancer activity [63].

**Figure 3 pharmaceutics-14-00070-f003:**
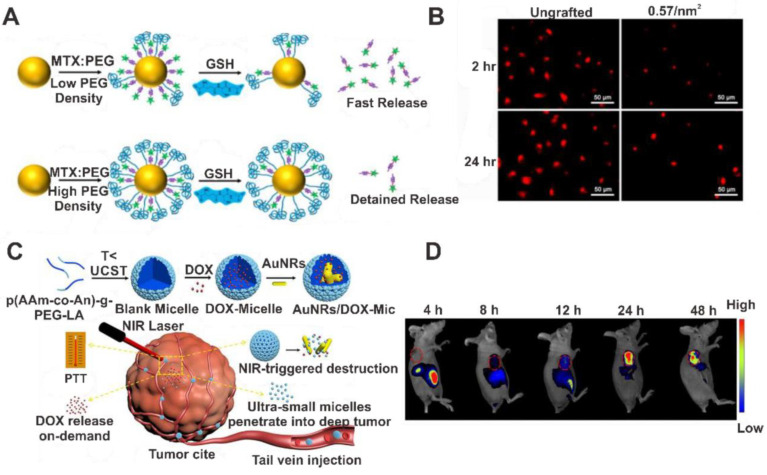
AuNP-polymer nanocomposites for use in drug delivery systems. (**A**) PEG density-dependent MTX release. (**B**) Live-cell fluorescent images of the MTX release from AuNPs of the samples with MTX (1 µM): PEG ratios of 1:0.1 or 1:0.05 at 2 and 24 h. (**C**) Light-responsive nanomicelles with AuNRs and doxorubicin co-loaded for synergetic photothermal-chemotherapy. (**D**) Fluorescence images of a tumor-bearing mouse after administration of AuNR/indocyanine green (ICG) micelles. Reproduced with permission from [54], Elsevier, 2016 and reproduced with permission from [62], American Chemical Society, 2021.

Studies showed that AuNPs with different sizes, morphologies, and surface modifications enhance the drug loading efficiency, and a comparative evaluation of drug loading efficiency using Au polymer hybrid nanocomposite, with a few recent literature reports, is described in Table 1. These studies showed that AuNPs with varying sizes, shapes, and different polymer coatings possess different drug loading efficiency and, depending on the mode of target action, one has to further optimize the conditions for efficient therapeutic applications due to the different biological aspects of cancer cells.

Thus, the use of Au and Au-polymer formulations resulted in smart drug delivery systems that minimized the side effects of cytotoxic anti-cancer drugs and enabled controlled drug release. The unique optical properties of AuNPs and the functionality of polymers are critical in these systems. Despite the significant progress, AuNPs cannot be easily replaced with organic carriers, such as natural polymers or liposomes, because of their potential toxicity compared with that of organic counterparts. Therefore, AuNPs should be used when the benefits of their use are much pronounced than the drawbacks.

### 3.2. Gene Delivery

Gene therapy is used to treat genetic ailments via the transfer of therapeutic genetic materials (e.g., deoxyribonucleic acid (DNA), messenger ribonucleic acid (mRNA), microRNA, small interfering RNA (siRNA), or short oligonucleotides) into cells to produce or suppress the production of specific proteins or correct specific genes using clustered regularly interspaced short palindromic repeats. Naked DNA or RNA molecules exhibit very short half-lives in biological fluids and do not easily permeate cell membranes, due to their negative charges. This limitation may be partially overcome using condensed nucleic acid structures, such as random or ordered aggregates of nucleic acids with or without template materials. A condensed structure of a nucleic acid may reduce the charge repulsion between the nucleic acid backbone and cell membrane by decreasing the surface’s negative charge, which may improve the uptake efficiency of the nucleic acid into the cell. Similar to cationic nanomaterials, AuNPs may act as gene carriers via the loading of DNA or RNA onto AuNP surfaces via covalent modifications, using thiol-modified nucleic acids or electrostatic interactions using positively charged AuNPs [69,70,71]. The challenges in gene delivery include the protection of therapeutic genes from the harsh conditions in the bloodstream, permeating the cell membrane, and structural integrity in the cytoplasm. These challenges were investigated using cationic polymeric materials (i.e., liposomes, dendrimers, and positively charged branched polymers) or self-assembled DNAs (DNA origami or nanocrystals) to protect the nucleic acids from enzymatic degradation and facilitate cell membrane translocation [72,73]. Lipofectamine^®^ and polyethyleneimine (PEI) are widely used materials for gene delivery owing to their excellent transfection efficiencies; however, their relatively high toxicity motivated extensive research for exploring novel materials and strategies in this field.

The use of AuNPs in gene delivery may yield several advantages that cannot be achieved using the polymeric counterparts, in terms of colloidal stability in the physiological state, the tunable density of DNA, and the performance of controlled DNA release via light stimulation.

The first example of the use of AuNPs in gene delivery was reported by Mirkin et al. in Science in 2006 [74]. AuNPs modified with oligonucleotides were proposed as intracellular gene regulation agents to control protein expression in cells. The oligonucleotide-modified AuNPs were less susceptible to nuclease degradation and exhibited a 99% cellular uptake efficiency, introducing oligonucleotides at a higher concentration than that of conventional transfection agents. The density of DNA on AuNPs could be controlled, which enabled tunable gene knockdown.

However, the loading amounts of nucleic acids on AuNPs were relatively low compared with those on polymeric carriers. This may be addressed using Au-polymer hybrid materials, such as the simple use of PEI-coated AuNPs, as proposed by Encabo-Berzosa et al. In this study, three different plasmid DNAs were bound to the NPs, with plasmids of ≤40 kb transfected, with superior performance compared to Lipofectamine^®^ [75]. Bishop et al. showed that layer-by-layer biodegradable polymer coatings containing two different polymers, 1-(3-aminopropyl)-4-methylpiperazine end-modified poly(*N*,*N*′-bis(acryloyl) cystamine-*co*-3-amino-1-propanol) containing disulfide bonds and 1-(3-aminopropyl)-4-methylpiperazine end-modified poly(1,4-butanediol diacrylate-*co*-4-amino-1-butanol), are poly(β-amino ester) containing ester linkages. Thus, these polymer coatings may be degraded in the cytoplasm via enzymatic hydrolysis and disulfide bond cleavage [76]. Cationic dendrimers combined with AuNPs were also used in RNA delivery [77,78,79]. Mbatha et al. reported folic acid (FA)-modified AuNPs mounted on poly(amidoamine) dendrimers. The sizes of the nano complex materials were optimal for bearing mRNA, and the polymer coating protected mRNA in the cellular environment against RNases [80].

Natural cationic polymers, such as chitosan (CS), were also used in gene delivery. Abrica-González et al. reported the use of CS-coated AuNPs as DNA carriers. AuNPs conjugated with CS, acylated CS, or CS oligosaccharides were used to evaluate the transfection efficiency of plasmid DNA in cell culture (HEK-293). The percentages of transfection obtained using CS, acylated CS, or CS oligosaccharides were 27%, 33%, and 60%, respectively. The positive charges formed on AuNP surfaces by the cationic amino groups of CS promoted the adsorption of plasmid DNA. The sizes of the AuNPs were 3–15 nm, and the higher the CS amount in the hybrid materials, the higher the plasmid DNA adsorption was, due to the positive charge [81].

Furthermore, controlled gene delivery was possible using AuNPs. An AuST-polymer hybrid structure was demonstrated as a controllable siRNA delivery system for use in cancer therapy [82]. The AuSTs absorbed the NIR light, even in deep tissues, thereby inducing a strong photothermal effect to release DNA and enhance hyperthermia.

Overall, Au or AuNP-polymer systems have been widely investigated as intelligent systems for use in gene delivery and exhibit promise for overcoming the existing issues in gene delivery systems.

### 3.3. Photoablation Therapy

Photothermal (PTT) and photodynamic therapies (PDT) utilize light energy to destroy abnormal cells [83]. PDT is an approved clinical therapeutic method, e.g., Visudyne^®^, for treating age-related macular degeneration. AuNPs and carbon materials, such as carbon nanotubes, were investigated as PTT agents owing to their strong absorption of light in the visible and NIR regions. Depending on their sizes and shapes, Au nanostructures exhibit considerably different light absorption and scattering properties in the visible and NIR regions (NIR-I (700–950 nm) and NIR-II (1000–1700 nm)), as shown in Figure 4 [84]. Furthermore, small NPs or NCs (<5 nm) are also applicable in PTT via in situ formations of large aggregates in cells. It was observed that the percentage increase in temperature depends on the NIR absorption coefficient of the NPs as well as on the power of the excitation source. Irradiation of NPs with NIR light increases the temperature of the medium, and it reaches a maximum value when the NIR absorption of AuNPs resonates with the laser wavelength. To attain this, AuNPs of different sizes and shapes are fabricated (Table 2). It was observed that different AuNPs possess different excitation wavelengths and unique light to heat conversion efficiency, from 22% to 103% depending on their size, shape, and aggregation states. As shown in Figure 4A, Au nanospheres and Au nanocubes possess a single surface plasmon absorption band, whereas nanorods, nanobranches, and nanobipyramids exhibit two major absorption bands. It is clear from Figure 3 there is a redshift in the absorption band from the nanosphere (Figure 4A, wave a) to the nanocube (Figure 4A, wave b). In the case of AuNRs depending on the aspect ratios, different absorption maxima are observed towards the red region in the spectra (Figure 4A, wave c–e). This suggests that AuNRs are a better candidate than AuNPs for PTT. Due to the longitudinal and transverse plasmon bands in AuNRs, they possess a reduced radiated attenuation effect, resulting in narrower linewidths compared to spherical nanoparticles, and possess strong NIR absorption efficiency, large scattering, and photothermal efficiency. Au nanobipyramid also showed redshift plasmon bands compared to AuNR, due to the strong electron oscillations in their conduction bands depending on the different aspect ratios (Figure 4B, wave a–d). The highest redshift and NIR absorption were observed for Au nanobranches, due to their considerable longitudinal electron oscillations (Figure 4B, wave e). Despite this being the case, it cannot be easily concluded which Au nanostructure would be best suited for particular cancer treatment, because of their respective advantages and features.

#### 3.3.1. PTT

PTT is based on the imperfect defense mechanisms of cancer cells against heat damage, which are inferior to those of normal cells. Cancer cells are highly susceptible to heat damage and, thus, PTT using anisotropic Au nanostructures has been extensively investigated for use in cancer therapy [95]. Ideal nanostructures for use in PTT should exhibit strong PT effects. Among various anisotropic structures, rod- or star-shaped or nanoshell structure have attracted considerable attention for their strong absorption in the NIR region, tunability, and relatively small sizes. Nanoshell structure has been investigated for the treatment of prostate and breast cancers in clinical trials [96].

Similar to that in drug delivery systems, polymer layer formation on Au nanostructures is required to improve their biological stability and achieve specific targeting. Accordingly, PEGylation or various targeting strategies have been widely investigated for use in PTT [97]. Shouju et al. showed that PEG-AuSTs exhibited high cell affinities and therapeutic efficiencies at tumor sites without targeting the normal cells [98]. Xu et al. reported Au nanocups with PEG surface modifications that exhibited PT efficiency and eliminated tumor tissues, both in vivo and in vitro [99]. To improve the PTT efficiency, a PEG-tethered AuNP-poly perylene-diimide semiconducting nanovesicle system was synthesized by Yang et al. [100]. The interparticle distance between the AuNPs and poly perylene derivative can be adjusted for better optical properties, and studies on U87MG tumor-bearing mice showed that these nanovesicles can act as excellent photoacoustic imaging agent. The in vivo PTT studies in tumor-bearing mice showed that laser irradiation of the tumor region resulted in complete suppression in tumor growth, as compared to bare AuNR and perlyenediimide. In vivo imaging and therapeutic evaluation demonstrated that the hybrid vesicle is an excellent probe for cancer theranostics.

Biocompatible polymers, besides PEG, have been widely studied to modify the surfaces of Au nanostructures to improve PT performance. Polypyrrole, -sarcosine (PS), -methoxyaniline, and -dopamine were investigated to stabilize AuNRs to improve their stability under irradiation. For example, polypyrrole is a conductive biomaterial recently used in surface coating and biomedical applications. Owing to its high conductivity and biocompatibility, the polypyrrole coating improves the biosafety and stability of AuNRs and enhances their PT conversion efficiency [101]. Wang et al. synthesized AuNRs with (poly-*o*-methoxyaniline) shells. The core-shell hybrid displayed an absorption band at 808 nm and an excellent PTT effect upon irradiation with an 808 nm NIR laser [102]. Owing to the conductive and electronic oscillation properties of the polymer, the poly-*o*-methoxyaniline shell not only improved the NIR absorption but also increased the photostability of AuNRs under NIR irradiation. Zhu et al. reported PS-modified AuNRs prepared via ligand exchange. The hybrid showed good stability in a wide pH range and under physiological conditions with the competitive ligand dithiothreitol. The Au-polymer hybrid exhibited an extended circulation time in the body, which was superior to that of PEG-AuNRs [103]. Pan et al. reported the use of Triton X-100 as an emulsifying agent in dopamine-conjugated AuNPs. Spherical Au nanostructures (106 nm) with 10 nm polydopamine shells were produced. The Au-polymer hybrid exhibited resistance to heat, acid, alkali, and irradiation at 808 nm, and an improved PT conversion efficiency (33%) compared to that of the uncoated AuNPs [104]. The formation of silica shell on AuNRs resulted in an improved PT conversion efficiency and improved stability of AuNR structure during light irradiation [105].

As spherical Au nanostructures only absorb light in the visible region, they are not widely investigated for use in PTT. However, various polymer materials may be used as templates for the assembly of spherical AuNPs with enhanced photostability and NIR absorption. For example, Wang et al. prepared a fibrous nanostructure using spherical AuNPs, with a silk fibroin (SF) template effect. The AuNP/SF nanofiber structure exhibited NIR absorption at 808 nm, with a higher PT conversion efficiency under an 808 nm laser than that of the non-assembled AuNPs (Figure 5A–C) [106]. In vitro and in vivo studies showed that the assembled AuNPs on the nanofibers efficiently killed breast cancer cells and destroyed breast cancer tumor tissues under PTT using a single NIR irradiation for 6 min, whereas the non-assembled AuNPs failed. The acridine orange/ethidium bromide (AO/EB) staining results showed that monodisperse AuNPs showed no dead cells (green fluorescence), whereas AuNPs/SF fiber showed 50% dead cells on irradiation (red fluorescence) with NIR laser, using 50 μg mL^−1^ of the nanomaterials.

Ultrasmall AuNCs (<5 nm) do not exhibit surface plasmon resonance effects. Therefore, the use of NCs in PTT has not been investigated. However, AuNCs exhibit strong luminescent properties and are sufficiently small for homogeneous distribution in the body [107,108]. These ultrasmall particles may accumulate easily in cancer cells and exhibit superior antitumor activities and homogeneous distribution in cell lines compared to those of their large analogs. The incorporation of indocyanine green (ICG) into Au_25_-glutathione leads to the effective transfer of ICG into tumor cells in vivo [109]. The ICG_4_-GS-Au_25_ (ICG_4_-glutathione-Au_25_NCs) enabled an effective NIR light-induced tumor PTT effect with complete ablation of the tumor cells. Conversely, the untargeted AuNCs dissociated in the liver and were removed via urine. This study highlights the role of integration of ICG in targeted PTT and the potentially fewer side effects of AuNCs compared to those of other AuNPs [110].

For improved therapeutic application, the PT effect may be combined with a drug delivery system for utilizing the synergistic effect of PTT and chemotherapy with drugs. Several studies using PTT and NIR-induced drug release systems have been reported. Docetaxel-loaded poly(lactide-*co*-glycolide) AuNRs [111] and diblock PEG-block-poly(2-hydroxyethyl acrylate) copolymer-coated AuNRs were proposed [112]. Xu et al. reported a dual-responsive drug-releasing nanocomposite containing hyaluronic acid-functionalized AuNRs that showed pH- and NIR-induced drug release. The nanocomposite exhibited a PTT-chemotherapy synergetic effect for treating breast cancer. The results of mouse experiments showed that the NPs exhibited an extended blood circulation efficiency, a high rate of accumulation in tumor cells, and excellent tumor suppression without toxicity over 20 days [113].

Tracking and quantifying internalized drugs in the various organs in the living system allow more detailed study into the efficiency of the nanocarrier system and further modifications in chemophotothermal therapy. Song et al. synthesized a gold nanoflower-based nanocarrier system for simultaneous DOX release, chemophotothermal therapy, and SERS imaging studies [114]. The gold nanoflower-like structure possesses high photothermal efficiency and hot spots needed for the SERS performance. Raman reporting molecule 4-mercapto benzoic acid was tagged with nanoflower, and further with cyclic peptide RGD, to obtain the targeted SERS imaging of A549 human lung cancer cells. After this, to impart stability, thiolated-polyacrylic acid was attached to the surface and DOX was loaded into the polymer via electrostatic interaction. Studies showed that maximum release of DOX was obtained at pH 5.3 and 37 °C under physiological conditions. SERs studies showed that the gold nanoflower nanocarriers entered the cytoplasm with a strong Raman signal intensity for the mercapto benzoic acid, and this incorporation study was further confirmed by the dark field microscopy studies. A chemophotothermal therapy study towards A549 human lung cancer cells showed that the nanocarrier system exhibited cell viability of 20% after NIR irradiation, indicating better antitumor efficiency than free DOX.

Recent studies reported diverse and improved therapeutic strategies designed based on the PTT effects of Au and Au-polymer systems. Although unlimited Au nanostructures with varying particles and complex geometry are available, ultrasmall NPs combined with polymers have also been designed for use in PTT. PTT combined with chemotherapy may improve the therapeutic effect, and controlled release is also possible. PTT may be further combined with PDT or immunotherapy.

#### 3.3.2. PDT

PDT is based on the generation of singlet oxygen and reactive oxygen species (ROS) upon light irradiation and is approved by the FDA for treating various diseases [115]. The critical component of PDT is a photosensitizer (PS) that generates ROS that is highly cytotoxic and leads to cell death via apoptosis. Typically, PSs are porphyrin derivatives, such as photofrin (630 nm), temoporfin (652 nm), verteporfin (690 nm), tookad (760 nm), and talaporfin sodium (chlorin e6 (Ce6) derivative, 667 nm) [115].

These drugs exhibit relatively low toxicity before light exposure but are cytotoxic when exposed to light. The non-selective activation and low solubility of PDT drugs are limitations that must be overcome. Therefore, diverse formulations for specific targeting and improved water solubility are widely investigated during PDT drug development. For example, Visudyne^®^ is formulated with liposomes and verteporfin, facilitating the localization of drugs in blood vessels of the retina.

Similar to anti-cancer drug delivery using AuNPs, AuNPs were used as carriers of PDT drugs for specific targeting to improve PDT [116,117,118]. The use of AuNPs as carriers of PDT drugs improves the therapeutic results via the PT effects of AuNPs and enhanced ROS generation. AuNRs may generate singlet oxygen, in addition to enhancing the photosensitization properties of neighboring PSs by facilitating energy transfer between donor-acceptor pairs [119]. For example, Li et al. prepared a two-photon excitation nanocomposite for use in PDT and imaging studies [120]. In this system, the two-photon absorption of tetraphenylporphyrin (TPP) is initially enhanced by the polymer backbone, and later by AuNRs. A silica coating acts as a spacer between the AuNR and the organic backbone to control the energy transfer. The two-photon excitation fluorescence and induced singlet oxygen generation are enhanced 792-fold when silica thickness is 9 nm. HeLa cells exhibited enhanced, bright fluorescence and effective PDT efficiency after irradiation of the NP system with an 800 nm laser.

The loading efficiency and solubility of PDT drugs can be improved using Au-polymers. For example, Kutsevol et al. prepared Au-polymer nanocomposites for use in PDT applications [119]. AuNPs and Ce6 were loaded into a star-shaped polymer-dextran-graft-polyacrylamide. The water solubility and pharmacokinetics of Ce6 are very poor, but incorporation into the Au-polymer coating enhanced its solubility and ROS generation properties. Even after prolonged heating, the sizes of AuNPs remained constant (10 nm), and PDT was conducted on the sub-lines of breast carcinoma MCF-7/S- sensitive to cytostatics, MCF-7/Dox- resistant to doxorubicin, and Jurkat cells’ suspension line of T-cell origin. After laser irradiation, cell deaths of 68.4%, 24.9%, and 20% were observed for the MCF-7/S, MCF-7/Dox, and Jurkat cells, respectively. The supramolecular three-component nanosystem displayed enhanced PDT results compared to those of the individual Ce6 dyes.

To address the circulation time in the blood, AuNPs were combined with zwitterionic polymers from the reticuloendothelial system; this combination resulted in an extended circulation time [121,122]. This type of Au-polymer hybrid may be used in combined PTT, PDT, and cell imaging studies. For example, Zheng et al. prepared a pH-responsive zwitterionic polymer which was grafted onto DOX-incorporating Au@TiO_2_ core-shell NPs. The acidic environment in cancer cells conferred a positive charge on the polymer backbone, which enhanced the cellular uptake and controlled the release of DOX at lower pH. Au@TiO_2_-core-shell exhibited enhanced PT and singlet oxygen generation properties, with the polymer material acting as a stabilizing agent and switching material for releasing DOX into human cervical cancer cell lines [123].

Rad et al. prepared functionalized acrylic copolymer NPs with spiropyran (SP) and imidazole groups via emulsion polymerization, and Au^3+^ was immobilized and reduced on the surface, yielding photoresponsive Au-bearing polymer NPs. The prepared nanocomposite was then further modified with folic acid (FA) as a site-specific tumor cell targeting agent. Microscopic studies showed the internalization of these nanomaterials in rat brain cancer cells (C6 glioma), with a 71.8% cellular uptake, in comparison with 28.8% for the nonconjugated materials. The nonpolar SP groups were converted to zwitterionic merocyanine isomers under UV irradiation at 365 nm, and their conjugation with AuNPs resulted in enhanced photogeneration of ROS. This was confirmed via intracellular ROS analysis and cytotoxicity studies using malignant C6 glioma cells. Owing to the strong surface plasmon resonance absorption of AuNPs, the FA-PAuNPs exhibit an elevated local PT efficiency under NIR irradiation at 808 nm. The prepared multifunctional FA-PAuNPs with comprehensive integration of prospective materials represent promising nanoprobes with targeting properties, enhanced tumor PDT, cell tracking, and PTT [124].

Liu et al. reported DOX loaded-polypeptide-based multilayer assembled AuNR (DH AuNR) for the combined chemotherapy and PDT study [125]. To reduce the toxic effect of CTAB coated AuNR and to improve the colloidal stability, the outer layer is coated with multiple layers of polymers. For the positively charged DOX loading, layer by layer coating of negatively charged polyglutamic acid (PGA) was used, and for further charge reversal, positively charged polylysine (PLL) was used. For targeting, the positively charged AuNR-PGA-PLL, nanoparticles were further modified with hyaluronic acid and helped to interact with the CD44 receptor overexpressed cancer cells with enhanced internalization. NIR laser irradiation-induced a burst release of DOX at pH = 7.4 and pH = 5, and the combination of DOX chemotherapy and AuNR photothermal effect resulted in 82.5% apoptosis in SKOV3 cancer cell lines, compared with free DOX and DH-AuNR without NIR (Figure 6A–C). The enhanced apoptosis rate is due to the generation of excessive reactive oxygen species from DH-AuNR. Overall, Au or Au-polymers have been widely investigated to improve PDT. Detailed studies may reveal novel Au-based smart nanomaterials for use in PDT and cell imaging applications.

### 3.4. PT Immunotherapy

Activation of the host immune system against cancer cells is the major objective of cancer immunotherapy. With considerable advances in this field, drugs such as monoclonal antibodies, dendritic cell-based vaccines, chimeric antigen receptor T cells, whole-cell vaccines, and immune checkpoint inhibitors were applied in clinical studies. However, inconsistent therapeutic responses of different patients, including organ toxicity and other side effects, are the major limitation [126]. In addition, cancer cells develop varied defense mechanisms to avoid recognition by the immune system via reducing surface antigen expression, inhibiting dendritic cells, and inducing apoptosis [127].

Eliminating primary tumors, activating the host immune system, and eliminating metastatic and residual tumor cells are critical for satisfactory cancer immunotherapy [3,128]. In this regard, the use of AuNPs is favorable because they may destroy primary tumors via strong PT effects induced by light stimulation, in addition to delivering immunostimulants or other drugs to enhance the antitumor immune responses of the body. Accordingly, combining local therapy with immunotherapy is a promising method to overcome the current limitations of immunotherapy. Furthermore, hyperthermia may cause dying tumor cells to release antigens, pro-inflammatory cytokines, and immunogenic intracellular substrates, thus promoting immune activation [129,130,131]. Therefore, the PTT effect achieved using AuNPs, represented by PT immunotherapy, may induce not only direct cell death but also immunogenic cell death (ICD).

However, applying the optimal thermal dose is critical for proper utilization of the synergistic effect of PTT and ICD [132]. Extremely high or low, a thermal dose may fail to properly induce ICD (Figure 7). This is one application of PTT in PT immunotherapy. Only the PTT effect is applied, without relying on any molecules to induce the immune response.

Although the tumor fragments produced via PTT can activate the anti-tumor immune response, it is insufficient. The combined use of immunostimulants or immunotherapy drugs can further improve PT immunotherapy. There are three different strategies, including combination with immunoadjuvants, immune checkpoint inhibitors, or chemotherapy.

#### 3.4.1. PT Immunotherapy with Immunoadjuvants

Zhou et al. synthesized AuNRs bearing mPEG-SH that, upon further surface modification with bovine serum albumin (BSA), resulted in negatively charged mPEG-AuNRs@BSA NPs. The immunoadjuvant R837 (imiquimod), which is positively charged, was adsorbed on the NPs via electrostatic adsorption. These immunoadjuvant-bearing AuNRs were used to treat melanoma via PTT and immunotherapy [133]. The nanocomposite increased the secretion levels of TNF-α and IL-6 and -12. The maturation level of dendritic cells was increased by 65.1%. The nanocomplexes under NIR irradiation effectively killed tumors and induced strong immune responses in treating metastatic melanoma in mice. Zhang et al. reported AuNSs conjugated with CpG oligonucleotide immunoadjuvants that acted as drug carriers, with PTT destroying cancer cells at the macro-level [134]. Immunotherapy studies showed that these nanomaterials were distributed inside the cell and killed cancer cells throughout the body. Chen et al. proposed the combination of PEI with AuNRs via S-Au bonds. This AuNR-PEI/CpG nanocomposite exhibited a PT response in vitro, and the immune system was activated following PTT, which was mainly attributed to the generation of tumor-specific antigens and the CpG adjuvant in situ. These findings indicate the potential use of cell-mediated nanoplatforms in tumor therapy via combining NIR PTT and immunotherapy [135].

#### 3.4.2. PT Immunotherapy with Immune Checkpoint Inhibitors

Immunotherapy with a specific immune checkpoint inhibitor is a promising method to disrupt the tumor immunosuppressive environment. This technique aims to block the overexpressing programmed cell death-1 (PD-1) in T cells and programmed cell death ligand-1 (PD-L1) in cancer cells, with the cytotoxic T lymphocyte antigen-4 (CTLA-4) protein receptors acting as checkpoints for cancer treatment. The PD-1/PD-L1 pathway mediates tumor immunosuppression by inhibiting T-cell proliferation and inducing T-cell apoptosis. To reverse tumor-mediated immunosuppression, anti-PD-1/PD-L1 antibodies were designed to block the PD-L1/PD-1 interaction.

The combination of AuNPs and inhibitors of PD-L1/PD-1 exhibited considerable potential in clinical applications [136]. The targeted delivery of immunoregulatory molecules using AuNPs may not only eliminate dead primary tumor tissues, but also promote a systemic immune response to treat metastatic lesions and prevent tumor recurrence [134]. Liu et al. reported plasmonic PEG-AuST-based NPs for use in PTT, in combination with immune checkpoint inhibitors of PD-L1 [137]. Complete eradication of primary treated and distant untreated tumors in several mice implanted with MB49 bladder cancer cells was observed. The cured mice received a re-challenge of MB49 cells and exhibited no new tumor formation after 60 days, which indicated the effectiveness of PT immunotherapy in MB49 cells. A multifunctional PVP-AuCG-based nanocarrier was developed by Cheng et al. to treat hepatocellular carcinoma [138]. The nanocarrier was loaded with ansamitocin P3 (AP3) and anti-PD-L1 antibodies (AP3-AuCGs-anti-PDL1), potentially combining PTT, chemotherapeutic agent-induced dendritic cell maturation, and checkpoint immunotherapy in one platform. Irradiation for 10 min using a NIR laser led to the formation of numerous dendritic cells with a significant release of AP3 and tumor-associated antigens, resulting in T-cell activation.

#### 3.4.3. PT Immunotherapy with Chemotherapy

Recent studies reported the improved performance of immunotherapy by combining PTT and chemotherapy. Nam et al. reported that combined PTT and chemotherapy induced potent anti-tumor immunity against disseminated tumors [139]. Polydopamine-coated spiky AuNPs were used as PT agents with extensive PT stability and efficiency. Remarkably, a single round of PTT combined with a sub-therapeutic dose of DOX elicited robust anti-tumor immune responses and eliminated local and untreated, distant tumors in >85% of animals with CT26 colon carcinoma. These studies provide novel strategies regarding a previously unrecognized, immunological feature of chemo-PTT and may help in developing novel therapeutic strategies against advanced cancer. Together, combination strategies involving AuNPs in the field of immunotherapy are promising therapeutic strategies for treating cancer.

### 3.5. Au Nanozymes for Cancer Treatment

The term nanozymes was coined by Paolo Scrimin’s group in the early 2000s [140] to represent the catalytic activity of AuNPs, such as their oxidase and peroxidase [141], glucose oxidase [142], and reductase [143] type activities. Recent studies showed that Au nanozymes can be used for biosensing and therapeutic applications. The excessive production of reactive oxygen species can damage cancer cells; Au nanozymes with peroxidase catalytic activities can induce hydrogen peroxide decomposition and ROS generation. This induced toxicity can be used for the PDT studies by inoculating the AuNPs to the specific tumor sites and activating their catalytic activity. For such actions, a low concentration of nanomaterial is needed, and this reduces the chances of unwanted generation of ROS in the other parts. The nanozyme activities are highly dependent on the size, shape, pH, temperature, and surface modification of AuNPs [144]. It was reported that by using NIR radiation in the therapeutic window, nanozymes can generate ROS for the PDT and PTT studies. Zhang et al. reported a carbon-gold hybrid nanozyme material for simultaneous real-time imaging, PDT, PTT, and nanozyme oxidase activities [145]. Mesoporous carbon nanospheres were doped with small AuNPs which were further complexed with reduced serum albumin and folic acid. Carbon mesoporous material contains several free carboxylic acid groups on the surface, which were further functionalized with serum albumin and folate receptor targeting folic acid to target the gastric tumor site via the 1-Ethyl-3-(3-dimethylaminopropyl)carbodiimide (EDC) coupling chemistry and NIR dye IR780 iodide via physical adsorption, which can act like a photothermal agent to enhance the photothermal efficiency and for cell imaging studies. Incorporation of AuNPs utilized as a nanozyme to catalyze the H_2_O_2_ located in the tumor cells can generate OH radicals for the intracellular oxidative damage. Irradiation of the hybrid material using a 808 nm laser resulted in enhanced PTT and PDT in MGC803 bearing mice. To confirm the nanozyme activity of AuNPs, control studies were performed using 2′,7′-dichlorodihydrofluorescein diacetate (H_2_DCFDA) dye, and it was observed that, in the absence of NIR light, certain green fluorescence was observed in the cell lines, indicating the nanozyme activity of AuNPs to produce the OH radical. This resulted in enhanced PDT under NIR light along with NIR dye IR780. In vivo experiments showed that the hybrid nanomaterial can suppress the tumor for up to 30 days, allowing the regeneration of new cells. The carbon mesoporous material enables the high drug loading efficiency of up to 33% for the improved PTT and PDT.

Combining one- or two-metal metallic NPs to enhance the biocatalytic activity is quite interesting in the nanozyme field. Gao et al. reported a dual nanozyme system containing ultrasmall Au and Fe_3_O_4_ NPs coloaded with dendritic mesoporous silica NPs, which were further functionalized with PEG for improved biocompatibility and physiological stability. The nanozyme accumulated in the tumor cell via the enhanced permeability and retention effect to activate the cascade catalytic reaction. AuNPs preserve the glucosidase activity to produce gluconic acid and H_2_O_2_, which are then catalyzed by the peroxidase-mimic Fe_3_O_4_ NPs to liberate high-toxic hydroxyl radicals (Figure 8) [146]. These OH radicals induced a tumor-suppression rate of 69% in both in vitro and in vivo studies. These hybrid nanozymes act like the natural Fenton type reaction to produce OH radical, so this hybrid material can be considered as a biomimetic system.

In another study, Fan, et al. reported that AuNPs encapsulated with hollow carbon shell nanospheres can be used for thermal cancer therapy along with the nanozyme activity under acidic conditions. The study showed that the ROS generation can be enhanced further with NIR laser irradiation in the therapeutic window, using a 808 nm laser to inhibit the CT26 tumor growth in vivo [147]. The simulation of the peroxidase and oxidase-like activities of Au nanozymes allows these nanozymes to treat tumors in human and animal cells with enhanced PTT and ROS generation. Recently, many research groups have studied the excellent therapeutic applications of nanozymes along with PTT and PDT activity, using various hybrid AuNP materials [148,149]. The oxidase- and peroxidase-like activity of AuNPs is the key feature for using nanozymes for cancer treatment. For example, studies performed by Maji et al. showed that graphene oxide-silicone-folic acid@AuNPs can impart intrinsic peroxidase catalytic activity and generate OH radicals, which can improve the therapeutic application of these nanozymes for cancer treatment in HeLa cells [150]. They showed that the nanohybrid system does not produce any OH radical in the normal tissues, so here, the folic acid receptor targeting action and acidity of the cancer cells play a key role in the nanozyme activity and subsequent cancer therapy. The hypoxia effect in solid tumors always renders the proper cancer treatment in radiotherapy. However, Yi et al. showed that the nanozyme activity of Au nanozyme@manganese dioxide nanoparticles could increase the oxygen in the radiotherapy-resistant hypoxia tumor tissue, despite reducing the toxic effects of NPs in other tissues. In another study, Liu et al. showed that the drug-resistance nature of cancer cells can be altered and anti-cancer drugs can be loaded into the tumor cell by using Polyamidoamine (PAMAM) dendrimer encapsulated AuNCs, which increase the consumption of H_2_O_2_ optimally by catalase to produce oxygen [151]. Quite interestingly, the nanozyme showed catalase-type activity even in highly acidic pH conditions (pH = 4). The enhanced amount of free amino groups in the PAMAM dendrimer become protonated in the acidic environment, which favors the adsorption of OH on the metal surface to trigger the catalyze type activity and generate ^1^O_2_ for PDT studies.

Therefore, as a part of the discussion, the various studies showed that owing to the higher biocompatibility of Au nanozymes, they can be used for various biosensing and therapeutic applications. The nanozymes can act like model complexes for the biological systems for mimicking various biochemical reactions. Most of the studies reported for Au nanozymes are in vitro, so in the near future, more in vivo studies can occur in this field to fully utilize the Au nanozymes for future therapeutic applications.

## 4. Tissue Engineering

Tissue engineering restores, replaces, or regenerates defective tissues with the aid of scaffolds. Various scaffolds for use in tissue engineering applications have been investigated. Cells, scaffolds, and growth factors are the three key components in tissue engineering. The normal cells in human tissues are partially anchored in a solid matrix represented by the extracellular matrix. This matrix provides structural support and mechanical properties and acts as a bioactive cue and reservoir for growth factors that mold the tissue during healing. The design and synthesis of the scaffold for use in tissue engineering aim to mimic the properties of the extracellular matrix [152,153].

The scaffolds should be biocompatible and biodegradable. In addition to the mechanical properties, porosity is also crucial. Polymeric and inorganic nanomaterials were used to enhance the scaffold properties. Among the various NPs, AuNPs have attracted considerable attention for their use in tissue engineering, owing to their diverse optical properties, simple surface chemistry, and rigidity. AuNPs may alter the mechanical properties of the scaffold and improve cell adhesion. This is critical, because cell adhesion and proliferation depend strongly on the elasticity and elastic moduli of the scaffolds. AuNPs (5 nm) incorporated into electrospun fiber consisting of polylactic acid resulted in an increase in Young’s modulus from ~30 to ~70 MPa, and the scaffold aided in the growth of skeletal muscle tissue [154]. AuNP incorporation may improve electrical conductivity and promote the propagation of electrical signals in cell-cell interactions, particularly in cardiac tissue engineering. The diverse roles of AuNPs in fabricating scaffolds for use in tissue engineering are shown in Figure 9 [155]. AuNPs of various sizes and shapes are incorporated into natural or synthetic polymer matrices to improve performance in various tissue engineering applications.

Nanofiber structures with incorporated AuNPs are frequently used in tissue engineering scaffolds owing to their high surface:volume ratios, structural similarities to extracellular matrices, and functions [156]. For example, Guterman et al. synthesized bacteria-inspired nanofibers that reduce metal ions. Using these nanofibers, AuNPs were prepared on a stable, conductive scaffold for use in tissue engineering [157]. While one method synthesizes AuNPs using the common citrate reduction or seed-mediated methods and incorporates them into the polymer nanofiber during electrospinning, another method uses the polymer in the in situ generations of AuNPs, and polymer selection depends on its ability to reduce Au^3+^ to Au^0^ and stabilize the formed AuNPs during the formation of the nanofiber scaffold [158].

Hydrogels are promising materials for use in tissue engineering owing to their tunable mechanical properties, injectability, and stimuli-responsive nature. Similar to the nanofibers and other scaffold structures, AuNP-hydrogel composites may yield novel advanced materials with good electrical and conductivity properties for use in tissue engineering. AuNPs were integrated into numerous hydrogel systems derived from alginate, gelatin, gelatin methacrylate, poly(*N*-isopropylacrylamide), and numerous other materials [159,160]. Such hybrid scaffolds may be generated either by preparing the hydrogel in a preformed NP solution or physically incorporating the AuNPs into the hydrogels. Grant et al. reported that an AuNP-containing hydrogel could be injected into swine ears, with the presence of AuNPs resulting in improved longevity of the material; in addition, the irritation level of the material was retained, suggesting that it could be used as a soft tissue filler [160].

Furthermore, the peptide sequence Arg-Glu-Asp-Val was incorporated in AuNPs, and the AuNPs were then used to produce hydrogel from alginate. This composite exhibited selective adhesion and proliferation of human umbilical vein endothelial cells. Additionally, the AuNP-alginate surface exhibited an improved cell adhesion rate and cell spreading compared to those of the bare alginate surface, which could be due to the interactions between AuNPs and vascular endothelial growth factor receptors on the plasma cell membrane. Moreover, this might also be due to the increased stiffness of the scaffold induced by AuNPs [161].

The uses of Au-polymer hydrogels in osteoblast cells are unlimited. Enhanced proliferation and growth rates were observed in preosteoblastic mouse cells cultured on chitosan/pectin thermosensitive hydrogels containing AuNPs [162]. Heo et al. reported that a photocurable gelatin hydrogel bearing AuNPs could differentiate human adipose-derived stem cells to osteoblast cell lines, with enhanced proliferation and growth rates [163]. This indicates that these hydrogels and their activities are comparable with those of the osteoinductive protein BMP-2, and they may thus be used as alternatives to BMP-2 in bone tissue engineering. AuNPs modified with *N*-acetyl cysteine were loaded into a gelatin hydrogel, and the proliferation and differentiation of human adipose-derived stem cells were studied in vitro and in vivo. The hydrogels loaded with AuNPs promoted the proliferation, differentiation, and alkaline phosphate activities of human adipose-derived stem cells, with dose-dependent differentiation towards osteoblast cells. Moreover, the in vivo results showed that these hydrogels loaded with high concentrations of AuNPs significantly influenced new bone formation [164].

The mechanical strength of scaffolds is critical for their application in tissue engineering. As their mechanical properties may affect the basic cell behavior, viability, proliferation, and adhesion, scaffolds with different elasticities and mechanical strengths may alter the rate of cell differentiation. Polymer materials may improve mechanical strength and stiffness to the AuNP-based scaffolds. For example, in a study by Baei et al., AuNPs with diameters of 7 nm were introduced into chitosan (CS) hydrogel. The AuNPs were formed within the CS gel before crosslinking, with the compressive modulus of the hydrogel increasing to 6.8 kPa. When mesenchymal stem cells (MSCs) were seeded within these scaffolds, a significant increase in the expression of the cardiac lineage differentiation markers Nkx-2.5 and α-MHC was observed, compared to that observed using plain CS hydrogels (Figure 10A,B) [165]. Hematoxylin and eosin (H & E) staining showed the even distribution of MSCs throughout the samples for both CS and CS-2AuNPs in their pores, and DAPI stain showed the homogenous pattern of MSCs after 14 days (Figure 10B). The role of Au in MSC cardiogenic differentiation was studied using immunohistochemistry staining for cardiac-specific markers α-MHC and Nkx-2.5; results showed the CS occasionally stained for the markers, whereas several α-MHC-/Nkx-2.5-positive cells were detected within the CS-2AuNP hydrogel. Both Nkx-2.5 and α-MHC proliferation rates were also increased, up to 24.3% and 35.3%, respectively, using CS-2AuNPs. Incorporation of large AuNPs with sizes of 60 nm into methacrylate-CS gel increased Young’s modulus of the scaffold from 450 Pa to 1.3 kPa, and these materials were used in cardiac tissue constructs [159]. Ahmed et al. prepared ε-polycaprolactone (ε-PCL) polymer nanofiber structures with AuNPs that showed good adhesion and proliferation properties for osteoblastic cells [166]. This indicates that the incorporation of AuNPs may increase the stiffness and mechanical strength of scaffolds. This increased stiffness is likely due to the electrostatic interactions between AuNPs and polymeric materials.

Electrically conductive scaffolds are promising platforms for use in tissue engineering. These are often used to form an electronic interface with the cells and enable facile stimulation of the tissues or recording of extracellular potentials [167,168]. This is particularly relevant for electrically active tissues, such as the cardiac muscle and neuronal tissues, wherein propagation of electrical signals is crucial for proper tissue function. For example, Nekounam et al. synthesized carbon nanofiber/AuNP-conductive scaffolds via electrospinning and electrospraying method for bone tissue engineering applications [169]. Polyacrylonitrile polymer solutions were integrated into conductive nanofibrous scaffolds and AuNPs were embedded either directly in the polymer solution or by co-electrospun with the polymer solution. The incorporation of AuNPs improved the flexibility of the process and the addition of 2.5% of AuNPs improved the conductivity of the material by 81%. This scaffold material exhibited no toxicity towards MG-63 cells, and SEM studies showed a uniform distribution of the cells on the scaffold with high proliferation rates [170]. Studies showed that incorporation of AuNPs to conductive materials can tune the stiffness and conductivity properties of the material with improved cell adhesion properties.

**Figure 10 pharmaceutics-14-00070-f010:**
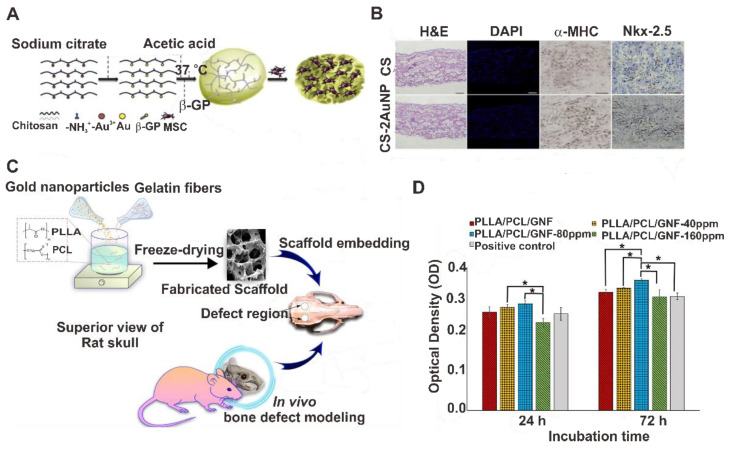
(**A**) Schematic illustration of the preparation and in vitro seeding of the CS-AuNP scaffold. (**B**) Morphology, distribution, and differentiation of MSCs within the CS-AuNP scaffolds. Hematoxylin and eosin and DAPI staining (scale bar: 200 μm) and immunohistochemistry analyses of Nkx-2.5 and α-MHC (scale bar: 50 μm) in a CS or CS-2AuNP hydrogel after 14 days of culture. (**C**) Schematic representation of the working principle of gelatin fiber scaffolds with AuNPs. (**D**) MG-63 cell proliferation on the synthesized gelatin fiber scaffolds with 24 h, 72 h incubation time, * *p* < 0.05. Reproduced with permission from [165], Elsevier, 2016; and [170], Nature Portfolio, 2021.

The engineered cardiac patches for treating damaged heart tissues are quite remarkable, and biocompatible 3D polymer materials are used for such purposes. However, these materials exhibit poor conductivities, resulting in their strong contraction as a unit. Au nanowires exhibit excellent conductivity, and their incorporation into biomaterials, such as alginate or polylactic acid, may yield scaffolds for use in cardiac tissue engineering. Dvir et al. showed that incorporation of Au nanowires into an alginate scaffold bridged the electrically resistant pore walls of alginate and improved electrical communication between adjacent cardiac cells. The tissues grown on these platforms are much thicker and more well-aligned than those grown on the pristine alginate scaffold. The major limitations of poor conductivity and irregular contractions can be minimized using this alginate-Au nanowire patches [25]. Recently, Li et al. showed that the incorporation of Au nanowires into a gelatin methacrylate hydrogel resulted in improved electrical conductivity and mechanical strength of the biomaterial scaffold [171]. The scaffold exhibited enhanced proliferation, and the studies of cardiomyocytes showed synchronous beating and faster spontaneous beating using the hybrid scaffold. This hybrid nanomaterial may be used for further applications in cardiac tissue engineering and drug screening.

Electrically conductive scaffolds are beneficial for engineering nerve tissues. In the field of nerve tissue engineering, nanofibrous Au-polymer scaffolds have shown promising results, as the AuNPs and their controlled physicochemical properties modulated the nerve signals due to the conductive cues developed by these nanocomposites. For example, poly-ε-caprolactone/chitosan (PCL/CS) mixtures with different proportions were electrospun to produce fiber structures, and citrate-reduced AuNPs were embedded in the fibers. Conductivity studies revealed that Schwann cells showed high conductance and biological responses on the AuNP-doped scaffolds, with a high proliferation rate. This indicates that the prepared AuNP-doped scaffolds may be used to promote peripheral nerve regeneration [172]. Kim et al. prepared a hybrid nanocomposite using Au and polyaniline that was studied as a nerve regeneration model [173]. Manipulation of the sizes, shapes, and conductivities of the hybrid materials was investigated in detail by varying the ratios of the components; the most effective conductive nanocomposite was delivered directly into the cells. Electroporation followed by mRNA-sequencing analysis of the conductive reinforced nanocomposite (CRNc)-internalized cells confirmed that these cells exhibited patterns similar to those of the positive group-induced neuron cells, with strongly expressed nerve fibers. Moreover, neural differentiation was studied by monitoring the growth of neurites from stem cells. The CRNc may be used to induce the formation of neuron-like cells by applying electrical stimulation to stem cells.

The use of AuNPs also resulted in an improved performance in bone tissue engineering. The formation of bone tissue is induced by the differentiation of MSCs toward osteoblasts and subsequent mineralization of the collagenous extracellular matrix of the bone. The addition of AuNPs into the cell culture media resulted in osteogenic differentiation of stem cells with increased mineralization [174,175]. Silica-coated AuNPs (Au@SiO_2_) were embedded in nanofibrous structures formed using biocompatible polymers. Poly-ε-caprolactone (PCL), PCL/silk fibroin (SF), and PCL/SF/Au@SiO_2_ nanofibrous scaffolds were prepared via electrospinning. Various microscopic studies showed that the addition of SF and Au@SiO_2_ enhanced the mechanical strengths, porosities, and adhesion capacities of the scaffolds for growing human MSCs into osteoblasts [176]. A 3D scaffold-based on gelatin nanofibers was synthesized by Samadian et al. The polylactic acid (PLLA)/PCL matrix polymer with gelatin nanofibers (GNF) was impregnated with AuNPs (Figure 10C). The design strategy of this nanocomposite is based on the following properties. The polymer materials PLLA/PCL act as matrix, and GNF mimic the bone extra cellular matrix, and AuNPs will act like a healing agent for the tissue engineering application as shown in Figure 10C. The addition of AuNPs increased the porosity of the material and maximum porosity was obtained for AuNPs with 160 ppm concentration. The porosity increased to 88.1 ± 2.16%, which suggested that this nanomaterial can be used for the tissue engineering applications. The LDH leakage assay showed that, PLA/PCL/GNF/AuNPs (160 ppm) scaffolds induced cytotoxic effect on cells through the damage to the cell membrane, but PLLA/PCL/GNF composite with 80 ppm AuNPs showed a higher proliferation rate, for MG-63 cells on the scaffold. Even though AuNPs at 160 ppm concentration showed higher porosity for the scaffold, the generation of ROS induced cell damage (Figure 10D). The resulting nanocomposite exhibited the highest neo-bone formation ability, osteocytes in the lacunae of the formed woven bone, and angiogenesis at the defect sites [170].

Thus, AuNP-polymer materials may be useful scaffolds for engineering various tissues. Along with the various advantages of AuNPs in improving mechanical strength, stiffness, and scaffold conductivity, additional functional groups on the polymers may further enhance the application of Au-polymer hybrid materials in nanomedicine.

## 5. Conclusions and Perspectives

Owing to the unique optical properties of Au, various novel applications and diverse strategies have been extensively investigated in therapeutics and tissue engineering. In this review, we focused on the recent progress in the use of Au and Au-polymer materials to overcome the current limitations in therapeutic and tissue engineering applications. Au nanostructures were used as carriers of drugs, such as anti-cancer drugs or PDT drugs, and therapeutic genes, as well. Au nanoparticles can also convert external light energy into thermal energy, which may be utilized to efficiently eliminate tumors. The PT-based method could be applied in cancer immunotherapy by improving the immune response using immunoadjuvants, immune checkpoint inhibitors, or chemotherapy. In the field of tissue engineering, AuNPs improved cell adhesion and the mechanical properties of polymer-based scaffolds; furthermore, Au can deliver external electrical stimuli to the cells, which could not be achieved by other systems. Accordingly, numerous novel applications of AuNPs and Au-polymer composites have been developed in therapy and tissue engineering. In spite of the improved therapeutic performances by use of Au or Au-polymer composites, it is expected that the clinical application of AuNPs will take time, until the potential toxicity issues are sufficiently addressed in the associated fields. The robust quality control of Au or Au-polymer systems is also a greatly important issue, in view of large scale production and to ensure reproducible therapeutic outcomes. Nevertheless, owing to the increasing number of clinical trials of novel formulations containing AuNPs, we expect that some of the composites will be clinically utilized in the near future.

## Figures and Tables

**Figure 1 pharmaceutics-14-00070-f001:**
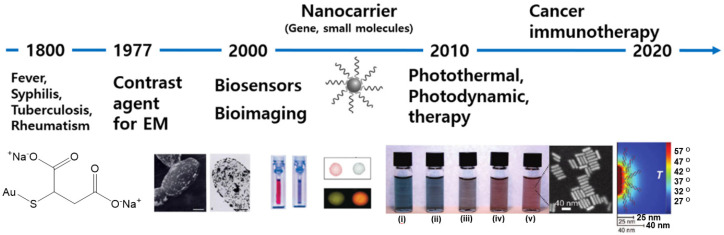
Timeline showing changes in the applications of AuNPs in the field of nanomedicine.

**Figure 2 pharmaceutics-14-00070-f002:**
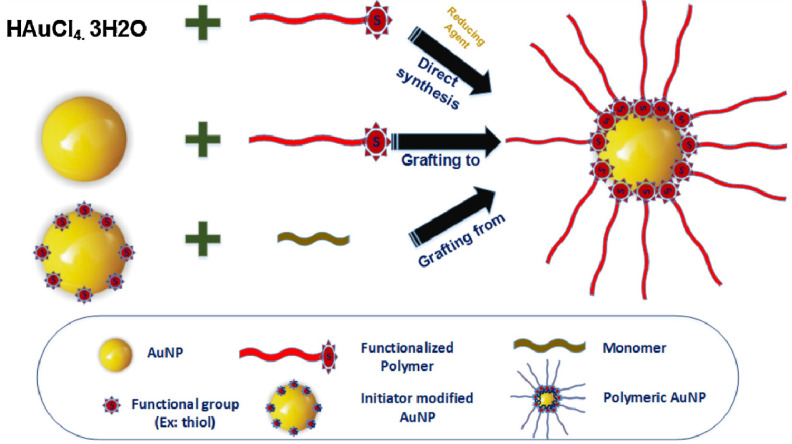
A schematic representing the various synthetic methods involved for polymer AuNPs. Reproduced with permission from [44]. Elsevier, 2015.

**Figure 4 pharmaceutics-14-00070-f004:**
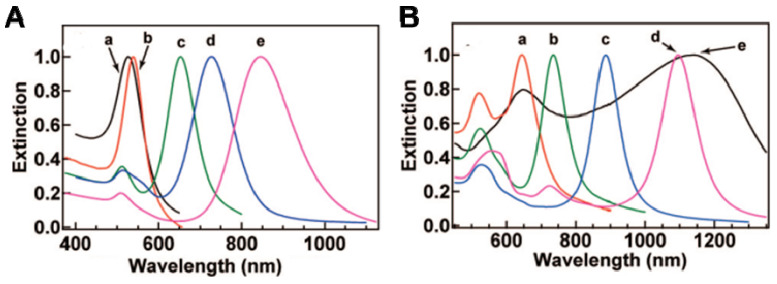
Extinction spectra of Au nanostructures. (**A**) Spectra of (a) nanospheres, (b) nanocubes, and nanorods with aspect ratios of (c) 2.4 ± 0.3, (d) 3.4 ± 0.5, and (e) 4.6 ± 0.8, respectively. (**B**) Spectra of nanobipyramids with aspect ratios of (a) 1.5 ± 0.3, (b) 2.7 ± 0.2, (c) 3.9 ± 0.2, and (d) 4.7 ± 0.2, and (e) nanobranches [84]. Reproduced with permission from [84]. American Chemical Society, 2008.

**Figure 5 pharmaceutics-14-00070-f005:**
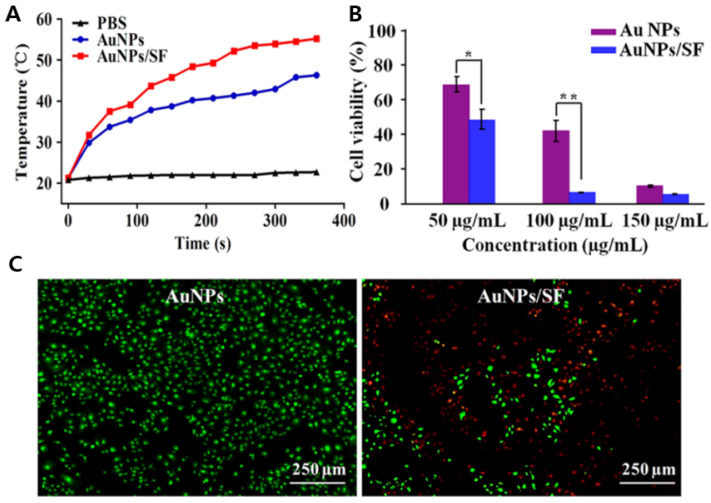
AuNP/silk fibroin hybrid (AuNP/SF) for use in PTT. (**A**) Temperature comparison using AuNPs and AuNP/SF. (**B**) Cell viability after AuNP or AuNP/SF uptake at various concentrations. (**C**) MCF-7 cell images before and after laser irradiation in the presence of AuNPs or AuNP/SF. Reproduced with permission from [106]. American Chemical Society, 2019.

**Figure 6 pharmaceutics-14-00070-f006:**
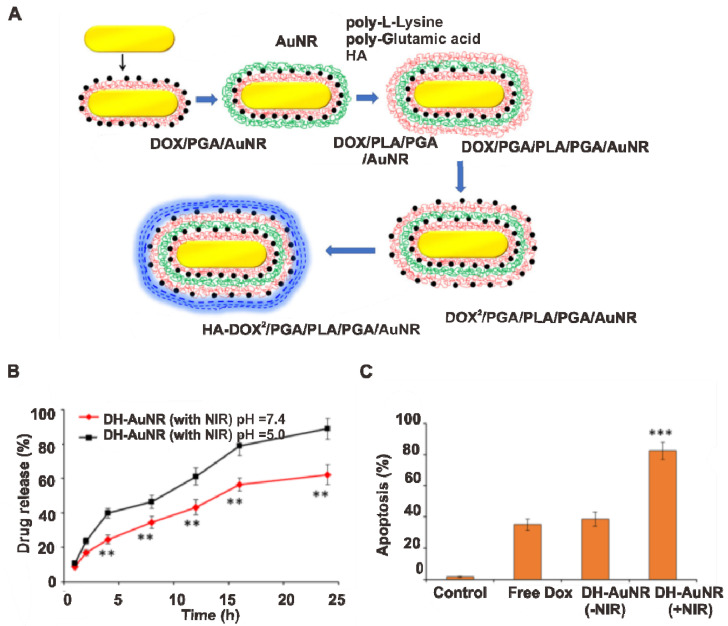
Polymer-AuNPs for use in PDT. (**A**) The schematic diagram for the layer-by-layer polyaminoacid coating on AuNR. (**B**) DOX release from the nanocomposite under different physiological pH, using NIR trigger. (**C**) Apoptosis analysis of SKOV3 cells after staining with Annexin-V and PI using flow cytometer and the results are represented as a bar diagram. and ** *p* < 0.001. *** *p* < 0.001. Adapted with permission from [125]. Portland Press Ltd., 2019.

**Figure 7 pharmaceutics-14-00070-f007:**
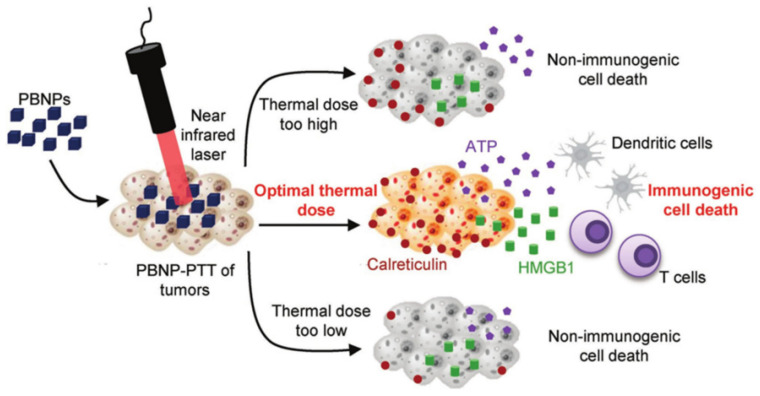
The required optimal thermal dose to induce ICD. Reproduced with permission from [132]. Wiley-VCH, 2018.

**Figure 8 pharmaceutics-14-00070-f008:**
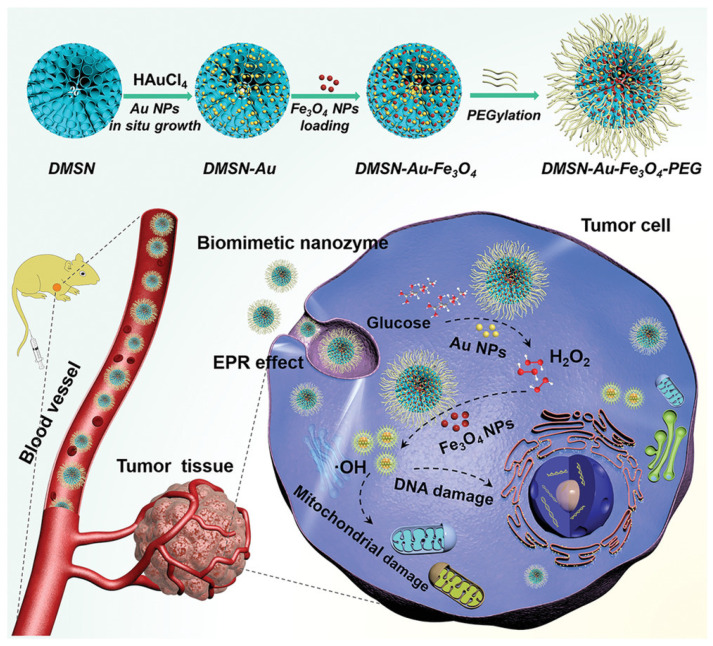
A schematic representation for the design and mode of action of the dual nanozyme system for the Fenton type reaction followed by its tumor ablation efficiency. Adapted with permission from [146].Wiley-VCH, 2018.

**Figure 9 pharmaceutics-14-00070-f009:**
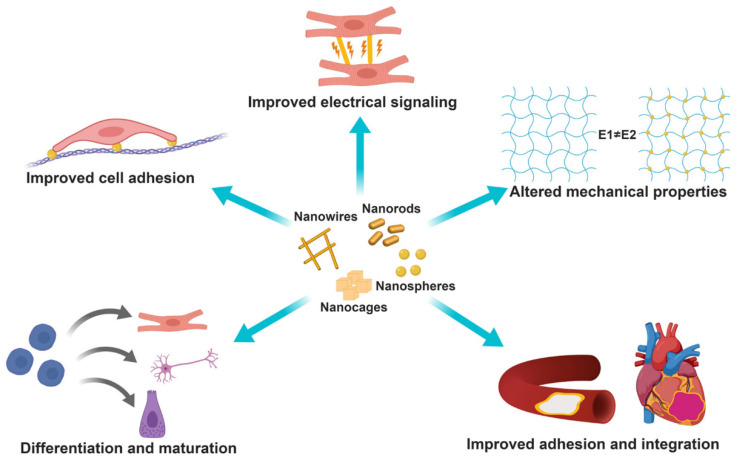
Schematic showing the uses of AuNPs in tissue engineering. AuNPs improve the mechanical strength, adhesion, and electrical and mechanical properties of the scaffold [155]. Reproduced with permission from [155]. American Chemical Society, 2019.

**Table 1 pharmaceutics-14-00070-t001:** Application of Au-polymer hybrid nanomaterials for drug delivery.

Modification of AuNP	Shape and Average Diameter (nm)	Drug and Drug Loading Capacity	Method of Functionalization	Cells Lines or Animal	Stability	Ref
PEG-folate	Spherical (120 nm)	Curcumin (1.3 ± 0.3 mg of Cur in 100 mg of the conjugate)	Hyaluronic acid (HA)- curcumin as linker	HeLa cells, C6 glioma cells, and Caco 2 cells	3 months at 20 °C	[64]
Polyacrylamide and folate	Spherical (30 nm)	MTX 43.8 ± 3.1 μg/mg of the nanogel	Covalent linking	KB cells	6 months	[65]
PEG	Spherical (155 nm)	Paclitaxel (36%)	Paclitaxel-PEG-SH as linker	HepG2 cells	Stable	[66]
PEG-carboxylic acids	Spherical (10 nm)	DOX (33%)	Covalent linking	HeLa cells	Stable	[57]
poly(acrylic acid)-b-poly(*N*- isopropylacrylamide)-b-poly (e-caprolactone)-SH	Hollow Nanostar (132 nm) and nanocage (238 nm)	DOX 78% in hollow nanostar and 84% using nanocage	Ionic, covalent, hydrogen bonding	MCF7 cells	2 months at 4 °C	[67]
Polydopamine and PEG	Nanorod (94 nm)	Methylene blue (42%) and DOX (52%)	Ionic and pi-pi interactions	HeLa cells	Stable	[68]

**Table 2 pharmaceutics-14-00070-t002:** Summary of photothermal conversion efficiency of gold nanoparticles for PTT and PDT. A correlation between size, morphology, and photothermal conversion efficiency.

Type of Gold	Size of Gold	Photothermal Efficiency	Laser	Treatment	Ref.
AuNR	17 × 56 nm	22%	0.4 W/cm^2^, 808 nm	PTT	[85]
AuNR	10 × 38 nm	95%	CW laser, 809 nm	PTT	[86]
AuCG	45 nm edge length, 5 nm wall thickness	64%	0.4 W/cm^2^, 808 nm	PTT/PDT	[87,88]
AuNP	20 nm	97–103%	0.28 CW laser, 532 nm	PTT	[89]
AuNSs	50 nm	59%	815 nm	PTT	[90]
AuNSs	152 nm	39%	CW laser, 2 W/cm^2^, 809 nm	PDT	[91]
AuNF	145 × 123 × 10 nm	74%	1 W/cm^2^, 809 nm	PTT	[92,93]
AuNRIs	400 nm	56%	CW laser, 0.1 W/mm^2^, 700–900 nm	PTT	[94]

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
