# Peer review of "Gold-Polymer Nanocomposites for Future Therapeutic and Tissue Engineering Applications"

_pharmaceutics, 2021, doi:10.3390/pharmaceutics14010070_

Round 1

Reviewer 1 Report

Lim summarized the application of Gold and gold-polymer composites in cancer therapy and tissue engineering. This review is well summarized and organized. However, before acception for publicatin, some minor issues should be solved.

  1. the title "Recent advances in the use of gold and gold-polymer nanocomposites in nanobiomedicine" is too big, as the antibacterial application of Au NPs also includes in nanobiomedicine.
  2. Line 143 “At s high temperature” there is a typo?
  3. If the methods to prepare the Au and Au-polymer hybrid was also included, the mascript will more completely.
  4. In the therapeutic applications section, I think the nanozyme therapy of Gold nanoparticle also falls into this category.
  5. To make a more comprehensive summary, some literatures should be added and discusssed, for example, Biomaterials, 2016, 100: 76-90; Journal of Materials Chemistry B, 2018, 6(19): 3030-3039, Theranostics, 2017, 7(8): 2177.

Author Response

Reviewer: 1

Lim summarized the application of Gold and gold-polymer composites in cancer therapy and tissue engineering. This review is well summarized and organized. However, before accepting for publication, some minor issues should be solved.

Comment 1. The title "Recent advances in the use of gold and gold-polymer nanocomposites in nanobiomedicine" is too big, as the antibacterial application of Au NPs also included in nanobiomedicine.

Response : We are thanking the reviewer for his valuable suggestion. We have changed the title of the Review as follow “Gold-polymer nanocomposites for future therapeutic and tissue engineering applications”.

Comment 2. Line 143 “At s high temperature” there is a typo?

Response:  We have removed this typo error in the modified review draft

Comment 3. If the methods to prepare the Au and Au-polymer hybrid were also included, the manuscript will be more complete.

Response: We have included a description in the modified review stating the synthetic protocol for AuNP and AuNP polymer materials. (Page 2, subheading 2 line number: 79-173).

Comment 4. In the therapeutic applications section, I think the nanozyme therapy of Gold nanoparticles also falls into this category.

Response: We have included this part in the revised review. (Subheading 2.5, Page number 11, Line number 742-825).

Comment 5. To make a more comprehensive summary, some literature should be added and discussed, for example, Biomaterials, 2016, 100: 76-90; Journal of Materials Chemistry B, 2018, 6(19): 3030-3039, Theranostics, 2017, 7(8): 2177.

Response:  We have included these references in the relevant places in the review

Reviewer 2 Report

This review covers the important topic of the combination of gold and gold-polymer nanocomposites and their use in nanobiomedicine. Indeed, this is an important area, gaining much interest in the recent years.

The manuscript is well organized and comprehensively described. However, the authors do not give perspectives on the current state of research and an outlook on where the future of the field is. Moreover, the author has only collated the literature and has not compared it critically.

 I would also suggest including a few tables. For example, in the section for therapy to illustrate:

  • Size and shape of the AuNP
  • Surface modification (polymer…)
  • Stability
  • Drug loaded and concentration
  • Animal or cell tested
  • Therapy

Comments about the reproducibility regarding the preparation of these materials should be also included.

Finally, the fonts in some figures are hard to read.

Author Response

Reviewer: 2

This review covers the important topic of the combination of gold and gold-polymer nanocomposites and their use in nanobiomedicine. Indeed, this is an important area, gaining much interest in recent years.

Comment 1. The manuscript is well organized and comprehensively described. However, the authors do not give perspectives on the current state of research and an outlook on where the future of the field is. Moreover, the author has only collated the literature and has not compared it critically.

Response : We have added a few lines in the conclusion and perspective part to highlight the importance of this research topic

Comment 2. I would also suggest including a few tables. For example, in the section for therapy to illustrate: Size and shape of the AuNP, Surface modification (polymer…), Stability, Drug loaded and concentration, Animal or cell tested, Therapy, Comments about the reproducibility regarding the preparation of these materials should be also included.

Response : Taking consideration of the reviewer's comment we have included an additional Table 2 in the drug delivery part to show the size, shape, and drug loading efficiency of a few polymer Au nanocomposites.

Comment 3. Finally, the fonts in some figures are hard to read.

Response : We have taken care of this issue in the revised review

Reviewer 3 Report

Dear,

The review proposed could be accepted in the journal after (minor) corrections and supplement information notably in the text of figure legend.

See also the suggested corrections in the pdf file

Author Response

Reviewer:3

Comment 1. The review proposed could be accepted in the journal after (minor) corrections and supplement information notably in the text or figure legend.

Response: We are thankful to the reviewer for considering our review for publication with minor corrections. The issues which were highlighted in the figure legends and comments are answered in the revised review.

Round 2

Reviewer 2 Report

The authors have addressed all concerns previously raised and this paper meets all requirements to be published in the present form.